# Practical Prediction Models of Tensile Strength and Reinforcement-Concrete Bond Strength of Low-Calcium Fly Ash Geopolymer Concrete

**DOI:** 10.3390/polym13060875

**Published:** 2021-03-12

**Authors:** Chenchen Luan, Qingyuan Wang, Fuhua Yang, Kuanyu Zhang, Nodir Utashev, Jinxin Dai, Xiaoshuang Shi

**Affiliations:** 1Key Laboratory of Deep Underground Science and Engineering (Ministry of Education), Department of Architecture and Environment, Sichuan University, Chengdu 610065, China; luanchenchen@stu.scu.edu.cn (C.L.); wangqy@scu.edu.cn (Q.W.); yangfuhua@stu.scu.edu.cn (F.Y.); 2018223055097@stu.scu.edu.cn (K.Z.); daijx@stu.scu.edu.cn (J.D.); 2Failure Mechanics and Engineering Disaster Prevention and Mitigation Key Lab of Sichuan Province, Chengdu 610065, China; nodir1987@stu.scu.edu.cn; 3Department of Mechanical Engineering, Chengdu University, Chengdu 610106, China

**Keywords:** low-calcium fly ash geopolymer concrete, tensile strength, reinforcement-concrete bond strength, regression analysis, cracking behaviors, design anchorage length

## Abstract

There have been a few attempts to develop prediction models of splitting tensile strength and reinforcement-concrete bond strength of FAGC (low-calcium fly ash geopolymer concrete), however, no model can be used as a design equation. Therefore, this paper aimed to provide practical prediction models. Using 115 test results for splitting tensile strength and 147 test results for bond strength from experiments and previous literature, considering the effect of size and shape on strength and structural factors on bond strength, this paper developed and verified updated prediction models and the 90% prediction intervals by regression analysis. The models can be used as design equations and applied for estimating the cracking behaviors and calculating the design anchorage length of reinforced FAGC beams. The strength models of PCC (Portland cement concrete) overestimate the splitting tensile strength and reinforcement-concrete bond strength of FAGC, so PCC’s models are not recommended as the design equations.

## 1. Introduction

### 1.1. Background

Low-calcium fly ash geopolymer concrete (FAGC) is regarded as an alternative to Portland cement concrete (PCC) [1,2,3] because of its low energy consumption and CO_2_ emissions, early compressive strength under slightly elevated temperatures [4], superior durability in sulphates or acids [5,6], and excellent fire-resistance behavior [7]. The necessary heat curing of FAGC is considered to limit its application in precast industries only, but the limitation can be overcome by pre-heating of fly ash and alkaline solution [8].

The most important application of concrete in building construction is concrete members [9]. Current research on structural behaviors of reinforced FAGC members focused on the failure mode and bearing capacity. However, there was a lack of study on other structural behaviors, such as the cracking behaviors and design anchorage length of reinforced FAGC beams, which are related to the tensile strength and reinforcement-concrete bond strength [9,10,11]. To estimate the cracking behaviors and calculate the design anchorage length, it is vital to accurately estimate the tensile strength and reinforcement-concrete bond strength from the compressive strength using the specific strength models of FAGC rather than the strength models of PCC.

There have been many studies to assess the tensile strength [12,13,14,15,16,17,18,19,20,21,22] and reinforcement-concrete bond strength of FAGC [12,13,14,23,24,25,26]. Some authors [12,13,14] argued that the splitting tensile strength of FAGC was higher than that of PCC at the same compressive strength, while others [15,16] obtained the opposite conclusion, which can be explained by the scatter of the splitting tensile strength of FAGC. The bond strength of FAGC was considered to be higher than that of PCC in the same case [12,13,14,23,24,25,26].

However, there were only a few attempts to develop the prediction models of tensile strength [15,16,21] and reinforcement-concrete bond strength [23,24], and there is still a long way to go before these prediction models can be used as design equations because of the limitations of these prediction models.

In previous attempts to develop the prediction models of the splitting tensile strength of FAGC, comparatively few data have been used considering the large scatter [15,16,21]. In addition, the specimens for strength test in different studies were different, but the effect of the shape and size of the tested specimens on strength was not considered in previous attempts [15,16,21]. Therefore, this work aims to provide a reliable tensile strength model by acquiring more data and considering the effect of shape and size of tested specimens on strength.

Bond strength should be regarded as a structural property rather than only a material property [27], however, in these previous attempts to develop the bond strength model, only the effect of compressive strength was considered, and the effects of structural factors, namely the cover *c*, bar diameter *d*, and development length *l*, on bond strength were ignored [23,24]. Therefore, these models cannot be used as design equations. Also, we noticed there was a lack of data on high-strength FAGC (compressive strength >50 MPa). In summary, for proposing design equations of bond strength of FAGC, this work considers the effects of structural factors and compressive strength on bond strength simultaneously, and acquires much data from these previous studies and the laboratory experiments.

### 1.2. Main Works

In this work, first, the splitting tensile strength and bond strength of four FAGCs with compressive strength of 50–80 MPa were tested, and previously published results of FAGC were collected to build databases; second, based on the databases, the prediction models of tensile strength and bond strength for FAGC, as well as the 90% prediction interval of the models, were proposed through regression analysis; third, the splitting tensile strength and bond strength of five FAGCs were tested and the results were used to validate the models; fourth, the models for FAGC were compared with the existing models for PCC to show whether the existing models are suitable for FAGC; finally, based on the strength models, the cracking behaviors of reinforced FAGC beams, such as the cracking moment, crack spacing and width, can be predicted using the existing prediction models of structural behaviors, and the design anchorage length can be calculated. The test results in previously published literature were used to validate the prediction models of the cracking moment and crack spacing.

### 1.3. Research Significance

At present, there is no design equation developed to predict tensile strength and reinforcement-concrete bond strength of FAGC as functions of compressive strength, which hinders the use of FAGC for large-scale field applications. Reliance on the existing design equation for PCC may lead to unsafe structural design. So far, a few studies have proposed prediction models for FAGC; however, there are limitations in those prediction models as shown in Section 1.1. This work addresses these knowledge gaps, provides practical prediction models of tensile strength and reinforcement-concrete bond strength of FAGC, and uses the strength models to estimate the cracking behaviors of reinforced FAGC beams and calculate the design anchorage length.

In addition, it notes that the data is from FAGC with different sources of fly ash and different mix designs, so the models do not depend on the sources of fly ash and the mix designs.

## 2. Materials and Methods

### 2.1. Strength Test

#### 2.1.1. Materials

Low-calcium fly ash (ASTM C618 Class F, Shenzhen Dot Co., Ltd., Shenzhen, China) was used as the source material for producing geopolymer binder. The oxide composition of fly ash as determined by X-ray fluorescence (XRF) (Shimadzu Corporation, Tokyo, Japan), is shown in Table 1. The particle size distribution of fly ash as determined by HELOS-RODOS (SYMPATEC GmbH, Clausthal-Zellerfeld, Germany), is shown in Figure 1.

The alkaline solution was a combination of sodium silicate (Na_2_SiO_3_, Foshan Zhongfa Silicate Co., Ltd., Foshan, China) with a modulus ratio (Ms) of 3.13 (where Ms = SiO_2_/Na_2_O, Na_2_O = 8.83%, SiO_2_ = 27.64%) and sodium hydroxide (NaOH, Chengdu Kelong Chemical Reagent Factory, Chengdu, China) prepared by dissolving NaOH solids (98% purity) in water. The alkaline solution was prepared approximately 24 h before usage.

Both coarse aggregates and fine aggregates in a saturated surface dry condition were used in this work. Crushed stone in three-grain sizes was used as coarse aggregate: 5–10 mm (50%), 10–16 mm (35%), and 16–20 mm (15%). Natural sand with fineness modulus of 2.3 was used as fine aggregate.

#### 2.1.2. Mix Proportions

The mix proportions of all mixtures are shown in Table 2. The tested results of Mix. 1–4 are added to databases for developing models, and those of Mix. 5–9 are for validating models. Their compressive strength is also given in Table 2. The compressive strength (*f_c_*) was tested on a 2000 KN electro-hydraulic mechanical testing machine (Changchun testing machine factory, Changchun, China) with a constant load rate of 3 KN/s using 100 mm cubes cured 28 days [28].

#### 2.1.3. Preparation of Test Specimens

The mixing of the concrete was undertaken in a compulsory mixer (Beijing Shenkeweijie Instrument and Equipment Co., Ltd, Beijing, China). First, fine and coarse aggregates were dry mixed in the mixer for 1 min, and then the fly ash was poured into the mixer for another minute. Second, the alkaline solution was sequentially added, and the mixing continued for 2 min. Finally, the fresh concrete was placed in steel molds, and they were then compacted on a table vibrator.

We cast 100 mm cubes for measuring the splitting tensile strength (*f_st_*) and modified direct pull-out specimens as shown in Figure 2 for measuring the bond strength (*τ*). Before casting, the steel bars were aligned with the center of the side baffles of the pull-out specimen molds and then fixed. Oil was applied to the inside surfaces of both the pull-out specimen molds and the cube molds. After casting, the specimens with the molds were covered with plastic film to prevent moisture loss, and they were then cured at 80 °C in an oven for 24 h. After removal from their molds, the specimens were cured in open air at 20–25 °C for 27 days.

#### 2.1.4. Splitting Tensile Strength Test

The tensile strength can be measured using the axial tensile strength test, splitting tensile strength test or flexural strength test. The axial tensile strength test is the most appropriate method to determine tensile strength, but it is more difficult than the splitting tensile strength test and the flexural strength test. Also, the value of the splitting tensile strength approaches that of the axial tensile strength, and it can also be used to predict the bond strength. Therefore, the splitting tensile strength test was used in this work. The splitting tensile strength was tested on a 300 KN capacity universal testing machine (Reger, Shenzhen, China) with a constant load rate of 300 N/s at ages of 28 days.

#### 2.1.5. Bond Test

Various types of specimens were used for the bond test. Popularly-used specimens include direct pull-out specimens and beam-end specimens. The direct pull-out specimens are easy to fabricate and test, but the surrounding concrete is in compression as the bars are in tension. This stress state differs from most reinforced concrete members, in which both the bars and the surrounding concrete are in tension. The beam-end specimens provide more realistic measures of bond strength in actual structures, but their fabrication and testing are complex. Therefore, we adopted a modified direct pull-out test, as shown in Figure 2 and Figure 3. The specimens were easy to fabricate and test, and the concrete remained in tension [29].

The specimen consisted of a concrete prism in which two HRB500 rebars of 16 mm diameter were embedded. The two bars were separated by 50 mm. When two bars were pulled in opposite directions, the bar with the shorter embedded length was pulled out. The shorter embedded length was 80 mm which was five times the bar diameter. The test was conducted using a 300 KN capacity universal testing machine with a constant displacement rate of 0.2 mm/min. The pull-out load was applied through the free ends of the embedded bars using friction grips.

The bond strength τ can be derived from Equation (1)
(1)τ=F/(πdl)
where *F* = ultimate force applied to bars; *d* = bar diameter, 16mm in this work; and *l* = embedded length of bar, 80mm in this work.

### 2.2. Database

A comprehensive literature review was conducted to establish databases for developing the models.

It is noted that different kinds of tested specimen for compressive strength and splitting tensile strength are adopted in different studies, so the effect of size and shape of tested specimens on strength is considered in this work. The common-used tested specimens include 100 mm cubes, 150 mm cubes, 100 × 200 mm^2^ cylinders (100 mm in diameter and 200 mm in length), and 150 × 300 mm^2^ cylinders (150 mm in diameter and 300 mm in length). The 150 × 300 mm^2^ cylinder is the standard specimen in ACI (American Concrete Institute) 318 and CEB-FIP (Fib, fédération internationale du béton), and the 150 × 300 mm^2^ cylinder strength is used in the models of ACI 318 and CEB-FIP. Therefore, to compare with the models of ACI 318 and CEB-FIP, the strength measured by the other specimens needs to be converted to that of the 150 × 300 mm^2^ cylinder.

The larger the volume of the concrete subjected to stress, the more likely it is to contain an element of a given low strength. As a result, the measured strength of the specimen decreases with an increase in its size. Due to the friction between the specimen and the testing machine, the measured compressive strength of the cube specimen is higher than that of the cylinder specimen with the length-to-diameter ratio of 2. So far, no study has investigated the effect of size and shape of tested specimens on strength for FAGC, but the above two conclusions still apply to FAGC. Therefore, in this work, we supposed that the conversion factors of compressive strength and splitting tensile strength for PCC also applied to FAGC. This assumption has also been used in reference [28].

The conversion factors of compressive strength and splitting tensile strength are shown in Table 3 and Table 4, respectively [30,31,32]. For example, for C60 concrete, according to Table 3, we can multiply the 100 mm cube compressive strength by the specific conversion factor 0.819 to obtain the 150 × 300 mm^2^ cylinder compressive strength.

### 2.3. Proposing Models and Corresponding 90% Prediction Intervals

The strength models were derived using the statistical regression analysis based on the databases. The least squares estimation procedure was used for the regression analysis. It uses the criterion that the solution must give the smallest possible sum of deviations of the observed $Y_{i}$ from the estimates of their true means provided by the solution [33].

Before solving with the models, the functions and the independent variables we selected were given, as shown in Equations (2)–(5). The existing models of PCC were used for reference, as shown in Equations (6)–(9). There were differences between ACI 318 and CEB-FIP, as follows.

Different functions were adopted.Besides the bar diameter to development length ratio (*d/l*) and the minimum cover to bar diameter ratio (*c_min_/d*), CEB-FIP also considered the bar diameter (1*/d*) and the maximum cover (*c_max_/c_min_*).

All the models were non-linear models, but Equations (2)–(4) can be linearized by an appropriate transformation on the dependent variable and there were explicit solutions. For nonlinear regression analysis like Equation (5), the iterative numerical method was used, where the values of the parameters in the existing model of PCC were used as the starting values of the parameters in the models of FAGC.
(2)$$f_{st}=kf_{c}^{n}$$
(3)$$f_{st}=aIn\left(f_{c}\right)-b$$
(4)$$\tau =k_{1}f_{c}^{n}{\left(\frac{1}{d}\right)}^{k_{2}}{\left(\frac{d}{l}\right)}^{k_{3}}{\left(\frac{c_{min}}{d}\right)}^{k_{4}}{\left(\frac{c_{max}}{c_{min}}\right)}^{k_{5}}$$
(5)$$\tau =\left(m_{1}+m_{2}\frac{c_{min}}{d}+m_{3}\frac{d}{l}\right)f_{c}^{n}$$

ACI 318 (America code) [34]:
(6)$$f_{st}=0.556f_{c}^{0.5}$$

ACI 318 [34]:(7)$$\tau =0.083\left(1.2+3\frac{c_{min}}{d}+50\frac{d}{l}\right)f_{c}^{0.5}$$

CEB-FIP (Europe code) [35]:(8)$$f_{st}=2.329In\left(f_{c}\right)-4.71$$

CEB-FIP [35]:(9)$$\tau =\frac{54}{4}{\left(\frac{f_{c}}{25}\right)}^{0.25}{\left(\frac{25}{d}\right)}^{0.2}{\left(\frac{d}{l}\right)}^{0.45}{\left(\frac{c_{min}}{d}\right)}^{0.25}{\left(\frac{c_{max}}{c_{min}}\right)}^{0.1}$$
where *d =* bar diameter, *l* = development length, *c_min_* = min{side cover, bottom cover, half of bar spacing}, *c_max_* = max{side cover, bottom cover, half of bar spacing}.

In addition, a data point may be an outlier because of errors in conducting the study (machine malfunction; recording, coding, or data entry errors; failure to follow the experimental protocol). Therefore, during modelling, we identified outliers through studentised residual $r_{t}^{*}$, as computed in Equations (10)–(12) [33]. When $r_{t}^{*}>3$, the data point can be regarded as an outlier.

After solving with the linear models, *t*-test was conducted to determine whether the dependent variable was related to each selected independent variable or not. The t-value, as computed in Equation (13), is compared to the corresponding critical value. In addition, the determination coefficient $R^{2}$, as computed in Equation (14), should be given to show the fitting precision of the models, which also applies to nonlinear models.

Note that these models only provide the estimated mean of the strength. Considering the large scatter of the strength of concrete, besides the models, the 90% prediction interval of the models, especially the lower limit of the prediction interval, is essential for design. The 90% prediction interval means there is a 90% probability that the true value of strength is within the interval. The lower limit of the 90% prediction interval is defined as the characteristic value, which means there is a 95% probability that the true value of strength is higher than the characteristic value. For linear models, the 90% prediction interval is given in Equations (18) and (19) which are derived from Equations (10) and (15)–(17) [33].
(10)$$s^{2}=\frac{\sum_{i=1}^{n}{\left(Y_{i}-\hat{Y_{i}}\right)}^{2}}{n-p-1}$$
(11)$$r_{i}=\frac{Y_{i}-\hat{Y_{i}}}{s\sqrt{1-v_{ii}}}$$
(12)$$r_{t}^{*}=r_{i}{\left(\frac{n-p-2}{n-p-1-r_{i}^{2}}\right)}^{0.5}$$
(13)$$t=\frac{\hat{\beta_{j}}}{s\sqrt{{\left(X^{\prime }X\right)}^{-1}}}$$
(14)$$R^{2}=1-\frac{\sum_{i=1}^{n}{\left(Y_{i}-\hat{Y_{i}}\right)}^{2}}{\sum_{i=1}^{n}{\left(Y_{i}-\bar{Y}\right)}^{2}}$$
(15)$$Y-\hat{Y}\;\sim \;N\;\left(0,\;\left[1+x^{\prime }{\left(X^{\prime }X\right)}^{-1}x\right]\sigma^{2}\right)$$
(16)$$P\left\{\left|\frac{Y-\hat{Y}}{\sqrt{1+x^{\prime }{\left(X^{\prime }X\right)}^{-1}x}s}\right|<t_{\alpha /2}\left(n-p-1\right)\right\}=1-\alpha$$
(17)$$P\left\{Y_{min}\leq Y\leq Y_{max}\right\}=90\%$$
(18)$$Y_{max}=\hat{Y}+t_{0.05}\left(n-p-1\right)\sqrt{\frac{\sum_{i=1}^{n}{\left(Y_{i}-\hat{Y_{i}}\right)}^{2}}{n-p-1}}\sqrt{1+x^{\prime }{\left(X^{\prime }X\right)}^{-1}x}$$
(19)$$Y_{min}=\hat{Y}-t_{0.05}\left(n-p-1\right)\sqrt{\frac{\sum_{i=1}^{n}{\left(Y_{i}-\hat{Y_{i}}\right)}^{2}}{n-p-1}}\sqrt{1+x^{\prime }{\left(X^{\prime }X\right)}^{-1}x}$$
where $\hat{\beta_{j}}$ is an unbiased estimator of the corresponding parameter of each independent variable; Y is the true value of the dependent variable; $\hat{Y}$ is an unbiased estimator of the mean of Y; $Y_{i}$ is each observed value of the dependent variable; $\hat{Y_{i}}$ is the prediction value of $Y_{i}$ by the model; $\bar{Y}$ is the mean of all observed values of the dependent variable; x is the 1×(p+1) matrix consisting of a column of ones, followed by the independent variables; X is the n×(p+1) matrix consisting of a column of ones, followed by the observations on independent variables; $v_{ii}$ is the *i*th diagonal element of $X{\left(X^{\prime }X\right)}^{-1}X^{\prime }$; $s^{2}$ is an unbiased estimator of $\sigma^{2}$; $\sigma^{2}$ is the variance of the random errors; n is the number of data points; p is the number of independent variables.

### 2.4. Validation of Models

Having developed the models in this research, the splitting tensile strength and bond strength of five FAGCs were measured according to Section 2.1.4 and Section 2.1.5, and then the results were used to validate the models. Their mix proportions and compressive strength are shown in Table 2.

## 3. Results

### 3.1. Database of Splitting Tensile Strength

A summary of the splitting tensile strength and corresponding compressive strength of FAGC is tabulated in Table 5, including 70 test results from previously published literature [14,15,16,17,18,19,20,21,22] and 20 test results from this work.

### 3.2. Database of Bond Strength

A summary of the bond strength, corresponding compressive strength, and size details of the specimens of FAGC are listed in Table 6, including 120 test results from previously published literature [12,13,14,23,24,25,26] and 12 test results from this work.

## 4. Discussion

### 4.1. Splitting Tensile Strength Model

Based on the basic equations as shown in Equations (2) and (3) and the database as listed in Table 5, models were proposed through simple linear regression analysis, as expressed in Equations (20)–(23) and shown in Table 7.

The *p* value < 0.01 shows that there is a highly significant linear relationship between the independent variable *X* and the dependent variable *Y*. Therefore, Equations (20)–(23) can describe the relationship between splitting tensile strength and compressive strength of FAGC satisfactorily. Although the resulting values of R^2^ in Table 7 do not seem sufficiently high, the R^2^ corresponding to the P value of 0.01 is only 0.07 when there are 90 data points [33].

According to *R*^2^, Equation (20) can describe the relationship better. In addition, when Equation (22) is adopted, a data point from reference [22] (*f_c_* = 86.41 MPa, *f_st_* = 7.43 MPa) is regarded as an outlier, while, when Equation (20) is adopted, no outlier is detected. It means that this data point is not a true outlier and Equation (22) is not a suitable model to express the relationship between splitting tensile strength and compressive strength. In summary, Equation (20), which uses the function in ACI 318, is recommended.
(24)$$f_{st,\;max}=0.2955f_{c}^{0.720}$$
(25)$$f_{st,\;min}=0.1393f_{c}^{0.724}\approx 0.686f_{st}$$

The 90% prediction interval of Equation (20) is given, as shown in Equations (24) and (25). It can be seen from Figure 4 that 92% of data points are in the prediction interval. If the value of splitting tensile strength needs to be given, the characteristic strength, the lower limit of the prediction interval, is recommended in consideration of the safety requirements of the structure. The lower limit is approximately 70% of the mean value. For PCC, CEB-FIP states that the lower limit of tensile strength is also 70% of the mean value [35]. This means the scatter of the tensile strength of FAGC is close to that of PCC.

Equations (21) and (23) show that the splitting tensile strength of FAGC is lower than that of PCC with the same compressive strength. The existing models of PCC overestimate the splitting tensile of FAGC, so the models of PCC cannot apply to FAGC.

### 4.2. Validation of Splitting Tensile Strength Model

The results of the splitting tensile strength and corresponding compressive strength of FAGC for validation of splitting tensile strength model is tabulated in Table 8, and plotted in Figure 5. It can be seen from Figure 5 that almost all data points are within the prediction intervals proposed by this work. Equation (20) is verified to predict the splitting tensile strength of FAGC.

### 4.3. Bond Strength Model

The bond strength model can be unified in the form of Equation (26), where *f(c, l, d)* shows the effects of structural factors on bond strength and *f_c_^n^* shows the effect of the material. For PCC, two different forms of functions *f(c, l, d)* were adopted in ACI 318 and CEB-FIP, as shown in Equations (7) and (9).
(26)$$\tau =f\left(c,l,d\right)f_{c}^{n}$$

During the modelling of Equation (4), 1*/d*, one of the dependent variables we selected, failed the *t*-test, while the other dependent variables passed the *t*-test. This means that the relationship between this dependent variable and bond strength is not significant. Therefore, this dependent variable was deleted, and then a new basic equation Equation (27) replaced Equation (4).
(27)$$\tau =f\left(c,l,d\right)f_{c}^{n}$$

Based on the database as listed in Table 6 and the basic equations as shown in Equations (27) and (5), Equations (28) and (29) were obtained to predict bond strength. According to the correlation coefficient (R^2^), Equations (28) and (29) give a good prediction of the bond strength, and Equation (28) is better. It is noted that these two equations are for unconfined FAGC specimens, as the effect of transverse reinforcement is not considered. However, considering that the transverse reinforcement is always beneficial to the bond strength, the bond strength predicted by Equations (28) and (29) is safe and can be applied to structural design. Figure 6 shows the comparison of the observation and prediction of bond strength by Equation (28).
(28)$$\tau =4.06f_{c}^{0.396}{\left(\frac{d}{l}\right)}^{0.456}{\left(\frac{c_{min}}{d}\right)}^{0.464}{\left(\frac{c_{max}}{c_{min}}\right)}^{0.341}\;R^{2}=0.632$$
(29)$$\tau =\left(0.249+0.269\frac{c_{min}}{d}+12.785\frac{d}{l}\right)f_{c}^{0.4}R^{2}=0.598$$

### 4.4. Comparison of Bond Strength of Portland Cement Concrete (PCC) and Fly Ash Geopolymer Concrete (FAGC)

In both Equations (28) and (29), the power of the compressive strength is near 0.4. The value is between 0.25 and 0.5, which means the contribution of compressive strength on the bond strength of FAGC is near that of PCC. Similarly to the bond strength of PCC, the bond strength of FAGC also increases with the increase of the cover to diameter ratio or the diameter to development length ratio.

However, the contribution of these structural factors on the bond strength of FAGC is different from that of PCC. We can divide Equation (28) by Equation (9) to obtain the relative value of the bond strength of FAGC to that of PCC in the same case. The result is given in Equation (30) and Figure 7.
$$\frac{\tau_{FAGC}}{\tau_{PCC}}=0.353f_{c}^{0.146}{\left(\frac{1}{d}\right)}^{-0.2}{\left(\frac{d}{l}\right)}^{0.006}{\left(\frac{c_{min}}{d}\right)}^{0.214}{\left(\frac{c_{max}}{c_{min}}\right)}^{0.241}$$
(30)$$=0.353f_{c}^{0.146}{\left(\frac{1}{d}\right)}^{0.008}{\left(\frac{1}{l}\right)}^{0.006}{\left(\frac{1}{c_{min}}\right)}^{0.027}c_{max}{}^{0.241}$$
$$\approx 0.353f_{c}^{0.146}{\left(\frac{1}{19}\right)}^{0.008}{\left(\frac{1}{550}\right)}^{0.006}{\left(\frac{1}{25}\right)}^{0.027}c_{max}{}^{0.241}=0.304f_{c}^{0.146}c_{max}{}^{0.241},$$

As shown in Equation (30), both the power values of the compressive strength and maximum cover are relatively large, so the relative value is mainly related to the compressive strength and maximum cover. The power values of the bar diameter, development length, minimum cover in Equation (33) are small, so their average values are selected to calculate their effect on the relative value. Bar diameters are commonly between 6–32 mm; the minimum cover is usually the bottom cover or the side cover, of size between 20–30 mm; the development length is usually between 100–1000 mm; the maximum cover is usually half of the spacing between bars, so it is usually between 20–100 mm; the compressive strength is usually 25–85 MPa. Therefore, the bond strength of FAGC is 1.00–1.76 times that of PCC in the same case, as shown in Figure 7. Note that in the case of lower compressive strength and a smaller maximum cover (i.e., a smaller spacing between bars), the bond strength of FAGC is only slightly higher than that of PCC.

In addition, it is noted that the above bond strength is the estimated mean value rather than the characteristic value. For the safety of structures, the characteristic value is more commonly used in structure design, as the probability that the true value of the strength is above the characteristic value is 95%, while the probability that it is above the estimated mean is only 50%.

Equation (31) in CEB-FIP gives the characteristic value of bond strength of PCC, which shows the characteristic value is 0.759 times the mean value for PCC [35]. In this work, the 90% prediction interval of Equation (28) is given, as shown in Equations (32) and (33). The lower limit of the prediction interval is the characteristic value of bond strength of FAGC, which shows the characteristic value is 0.616 times the mean value for FAGC. This means that the scatter of bond strength of FAGC is larger than that of PCC. Therefore, it is important to compare the characteristic values of bond strength of PCC and FAGC after comparing the mean values of the bond strength of PCC and FAGC.

The comparison of the characteristic values of bond strength of PCC and FAGC is shown in Equation (34) obtained through dividing Equation (33) by Equation (31). The result is plotted in Figure 8.

Previous literature [12,13,14,23,24,25,26] showed that the bond strength of FAGC is higher than that of PCC with the same case, so it was concluded that using the PCC model to predict the bond strength of FAGC is conservative but feasible. However, this work shows that although the mean value of bond strength of FAGC is higher than that of PCC in the same case, the characteristic value of bond strength of FAGC is lower than that of PCC in the case of a smaller maximum cover (i.e., a smaller spacing between bars). This means that the prediction model of bond strength for PCC cannot be used directly for FAGC.
(31)$$\tau_{PCC,c}=\frac{41}{4}{\left(\frac{f_{c}}{25}\right)}^{0.25}{\left(\frac{25}{d}\right)}^{0.2}{\left(\frac{d}{l}\right)}^{0.45}{\left(\frac{c_{min}}{d}\right)}^{0.25}{\left(\frac{c_{max}}{c_{min}}\right)}^{0.1},$$
$$\tau_{FAGC,max}=6.153f_{c}^{0.397}{\left(\frac{d}{l}\right)}^{0.440}{\left(\frac{c_{min}}{d}\right)}^{0.469}{\left(\frac{c_{max}}{c_{min}}\right)}^{0.349}$$
(32)$$=1.5155f_{c}^{0.001}{\left(\frac{d}{l}\right)}^{-0.016}{\left(\frac{c_{min}}{d}\right)}^{0.005}{\left(\frac{c_{max}}{c_{min}}\right)}^{0.008}\tau$$
$$\approx 1.5155{\left(55\right)}_{}^{0.001}{\left(\frac{19}{550}\right)}^{-0.016}{\left(\frac{25}{19}\right)}^{0.005}{\left(\frac{60}{25}\right)}^{0.008}\tau =1.629\tau ,$$
$$\tau_{FAGC,c}=\tau_{FAGC,min}=2.672f_{c}^{0.395}{\left(\frac{d}{l}\right)}^{0.472}{\left(\frac{c_{min}}{d}\right)}^{0.459}{\left(\frac{c_{max}}{c_{min}}\right)}^{0.332}$$
(33)$$=0.658f_{c}^{-0.001}{\left(\frac{d}{l}\right)}^{0.016}{\left(\frac{c_{min}}{d}\right)}^{-0.005}{\left(\frac{c_{max}}{c_{min}}\right)}^{-0.008}\tau$$
$$\approx 0.658{\left(55\right)}_{}^{-0.001}{\left(\frac{19}{550}\right)}^{0.016}{\left(\frac{25}{19}\right)}^{-0.005}{\left(\frac{60}{25}\right)}^{-0.008}\tau =0.616\tau ,$$
$$\frac{\tau_{FAGC,c}}{\tau_{PCC,c}}=0.306f_{c}^{0.145}{\left(\frac{1}{d}\right)}^{-0.2}{\left(\frac{d}{l}\right)}^{0.022}{\left(\frac{c_{min}}{d}\right)}^{0.209}{\left(\frac{c_{max}}{c_{min}}\right)}^{0.232}$$
(34)$$=0.306f_{c}^{0.145}{\left(\frac{1}{d}\right)}^{-0.013}{\left(\frac{1}{l}\right)}^{0.022}{\left(\frac{1}{c_{min}}\right)}^{0.023}c_{max}{}^{0.232}$$
$$\approx 0.306f_{c}^{0.145}{\left(\frac{1}{19}\right)}^{-0.013}{\left(\frac{1}{550}\right)}^{0.022}{\left(\frac{1}{25}\right)}^{0.023}c_{max}{}^{0.232}=0.257f_{c}^{0.145}c_{max}{}^{0.232},.$$

### 4.5. Validation of Bond Strength Model

The results of the bond strength for validation of the bond strength model and the corresponding compressive strength and size details of the specimens of FAGC are listed in Table 9.

It can be seen from Figure 9 that all data points are within the prediction interval proposed by this work. The bond strength model is verified. Meanwhile, it should be noted that all data points are below the estimated mean value, which shows the importance of the characteristic value in structural design.

### 4.6. Design Anchorage Length of Reinforced FAGC Beams

The design anchorage length *l_b_* may be calculated from Equation (35).
(35)$$l_{b}=\frac{f_{y}d}{4\tau }$$
where *f_y_* = tensile strength of the reinforcement.

To ensure reliable anchorage lengths, the characteristic value of bond strength as shown in Equation (33) is adopted. Then, the calculation equation of design anchorage length of reinforced FAGC beams is derived, as shown in Equation (36).
(36)$$l_{b}={\left(0.0936f_{y}f_{c}^{-0.395}d^{0.987}c_{min}^{-0.127}c_{max}^{-0.332}\right)}^{1.894}$$

The curves of the design anchorage length of reinforced FAGC beams vs. the maximum cover for different cases are calculated based on Equation (36) and compared with those of reinforced PCC beams proposed by GB 50010–2010 [36], as shown in Figure 10. The design anchorage lengths of reinforced FAGC beams are much longer than those of reinforced PCC beams in the case of a small maximum cover (i.e., a smaller spacing between bars). This can be explained by the fact that the bond strength of FAGC is lower than that of PCC in this case, as shown in Figure 8. Based on this, to obtain suitable design anchorage lengths of reinforced FAGC beams, the minimum bar spacing needs to be restricted in the design code for FAGC.

### 4.7. Prediction of Cracking Behaviors of Reinforced FAGC Beams

The cracking moment of reinforced concrete beams is much smaller than the ultimate bearing capacity, so most of them are in the cracking state, and the cracking behaviors, such as cracking moment and crack spacing and width, deserve our attention.

#### 4.7.1. Cracking Moment

The cracking moment *M_cr_* of a rectangular cross-section beam may be predicted from Equation (37).
(37)$$M_{cr}=0.256f_{t}bh^{2}$$
where *f_t_* = concrete tensile strength, *b* = width of rectangular cross-section, *h* = height of rectangular cross-section.

By incorporating the tensile strength model of FAGC proposed by this work (Equation (20)) to Equation (37), the cracking moment can be predicted based on the compressive strength and Equation (38). To validate the prediction model, the experiment results on the cracking moment of reinforced FAGC beams were collected from previous literature [37], as listed in Table 10. The corresponding *M_cr_* and test/prediction ratios predicted by Equation (38) are also shown in Table 10. The average test/prediction ratio of *M_cr_* is 1.00, with a standard deviation of 0.06. *M_cr_* predicted by Equation (38) is closer to the test result than *M_cr_* predicted by reference [37]. Both indicate that Equation (38) can predict *M_cr_* of reinforced FAGC beams well.
(38)$$M_{cr}=0.256\left(0.203f_{c}^{0.722}\right)bh^{2}$$

#### 4.7.2. Crack Spacing and Width

For reinforced PCC beams, the average crack spacing *S_cr_* can be analytically evaluated by many models in papers or codes. The results of Khuram Rashid et al. showed that incorporating the rebar-concrete bond strength in Zhang’s model resulted in more precise analytical values than using the design node of JSCE (Japan Society of Civil Engineers) or CEB-FIP for blended slag and fly ash geopolymer concrete [38]. However, they still used the bond strength model for PCC.

In this work, the crack spacing was evaluated by using Zhang’s model as shown in Equation (39) [39]. And to obtain a precise result for FAGC, the bond strength model used Equation (28) and the tensile strength model used Equation (20). See reference [39] for the calculation of *A_t_*. *l* in Equation (28) should be *S_cr_*, as the crack spacing is the development length of concrete through which concrete stress grows from zero to tensile strength. Then, the prediction model of the crack spacing of reinforced FAGC beams was obtained, as shown in Equation (40). 

Zhang’s model [39]:
(39)$$\mathrm{Zhang}’s\;\mathrm{model}\;[39]:\;S_{cr}=1.5\frac{f_{t}A_{t}}{\tau \sum^{\;}O_{r}}$$
where $\sum^{\;}O_{r}$ = total perimeter of tension reinforcement, *A_t_* = entire effective tension area. 

To validate the prediction model, the experimental results on the crack spacing of reinforced FAGC beams were collected from previous literature [37,40], as listed in Table 11. Based on the details of these tested reinforced FAGC beams as shown in Table 12, the corresponding *S_cr_* and test/prediction ratios can be predicted by Equation (40), as shown in Table 11. The average test/prediction ratio of *S_cr_* is 1.11, with a standard deviation of 0.17. This indicates that Equation (40) can predict *S_cr_* of reinforced FAGC beams well.
(40)$$S_{cr}={\left(1.5\frac{0.203f_{c}^{0.326}A_{t}}{4.06d^{0.456}{\left(\frac{c_{min}}{d}\right)}^{0.464}{\left(\frac{c_{max}}{c_{min}}\right)}^{0.341}\sum^{\;}O_{r}}\right)}^{1/0.544}$$

The crack width *w* can be obtained from the crack spacing, as it is the tensile extension difference between the reinforcement and the concrete within the crack spacing [41]. Considering that the tensile strain of concrete is very small, the crack width can be derived from Equation (41).
(41)$$w=\left(\frac{\sigma_{s}}{E_{s}}\right)S_{cr}$$
where *σ_s_* = stress in the reinforcement at a cracked section, *E_s_* = modulus of elasticity of the reinforcement.

Based on Equations (39) and (41), it is concluded that from a qualitative perspective, compared with PCC, the higher bond strength and the lower tensile strength of FAGC leads to the narrower crack width of reinforced concrete beams, which is beneficial to the durability of reinforced concrete beams.

## 5. Conclusions

This study has established the databases of splitting tensile strength and bond strength of FAGC, developed and verified the prediction models and the corresponding prediction intervals of splitting tensile strength and bond strength of FAGC, and used the strength models to calculate the design anchorage length and estimate the cracking moment, crack spacing and width of reinforced FAGC beams.

Based on this study, the following conclusions are obtained:Compared with the previous strength models of FAGC, the tensile strength model in this study considers the effect of shape and size of tested specimens on strength, and the bond strength model in this study considers the cover to diameter ratio and the diameter to development length ratio. Therefore, the models in this study can be used as the design equations for estimating the tensile strength and reinforcement-concrete bond strength of FAGC.The strength models provide the corresponding 90% prediction intervals. The lower limit of the prediction intervals is the characteristic value of the strength.The splitting tensile strength of FAGC is slightly lower than that of PCC with the same compressive strength, while the scatter of the splitting tensile strength of FAGC is close to that of PCC.The scatter of the bond strength of FAGC is larger than that of PCC. This results in the fact that for the bond strength of FAGC, although the estimated mean value is higher than that of PCC in the same case, the characteristic value may be lower than that of PCC in the case of a small bar spacing.The strength prediction models of PCC cannot be used for FAGC.To ensure adequate anchorage and suitable design anchorage lengths of reinforced FAGC beams, the minimum bar spacing needs to be restricted in the design code for FAGC.Incorporating the models into the prediction models of the cracking behaviors for PCC gives good predictions on the cracking moment and crack spacing of reinforced FAGC beams.

## Figures and Tables

**Figure 1 polymers-13-00875-f001:**
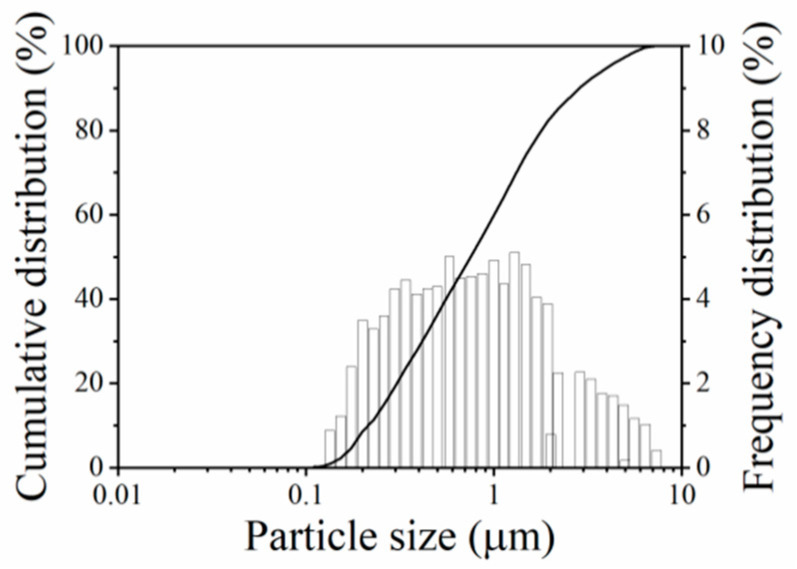
The particle size distribution of fly ash.

**Figure 2 polymers-13-00875-f002:**
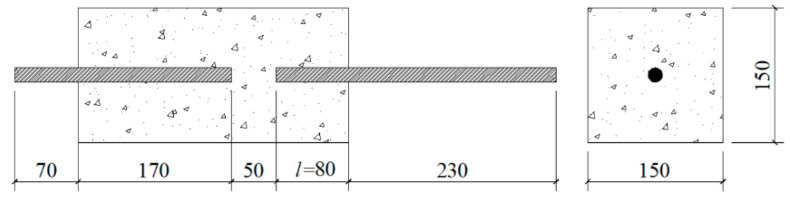
Detail of the modified direct pull-out specimen (mm) (*l* = the shorter embedded length of the bar).

**Figure 3 polymers-13-00875-f003:**
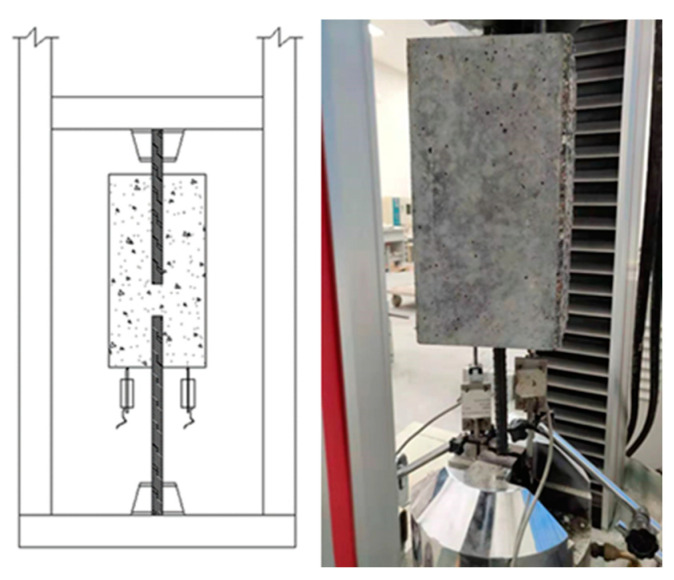
Test set up for the bond test.

**Figure 4 polymers-13-00875-f004:**
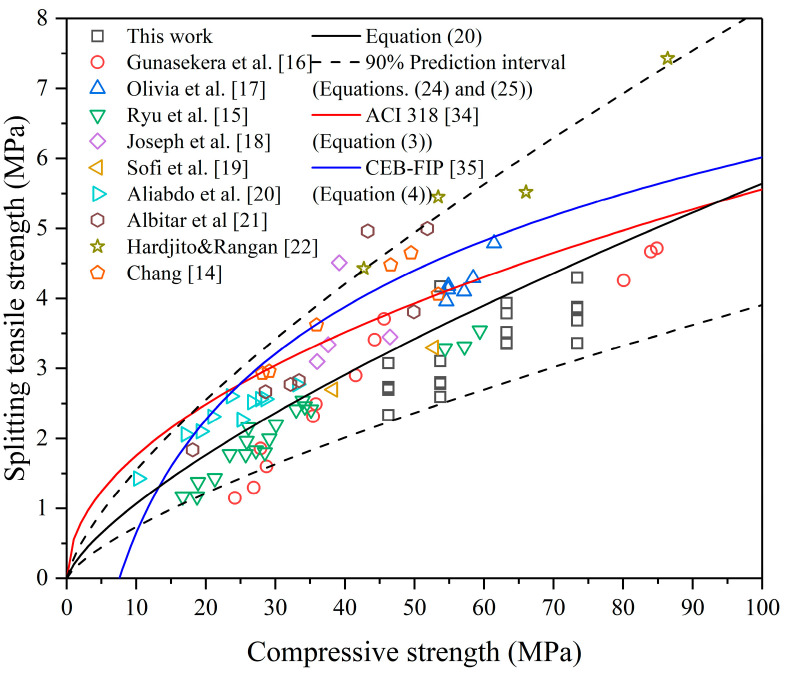
Relationship between the splitting tensile and compressive strength of FAGC.

**Figure 5 polymers-13-00875-f005:**
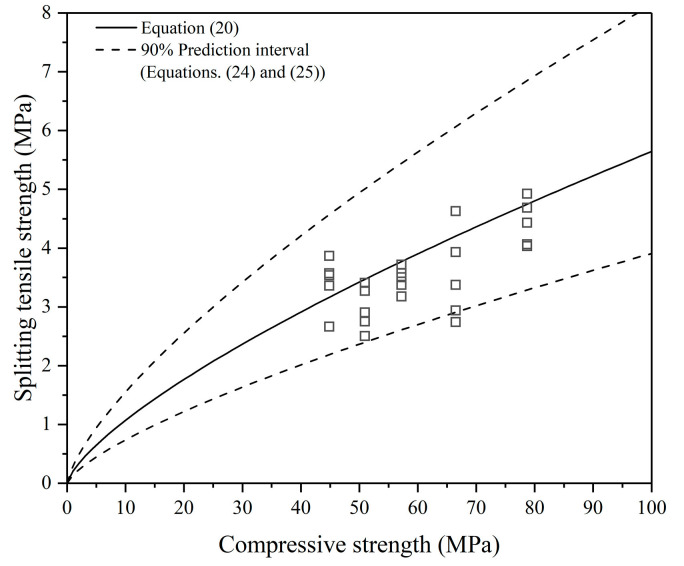
Validation of the splitting tensile strength model.

**Figure 6 polymers-13-00875-f006:**
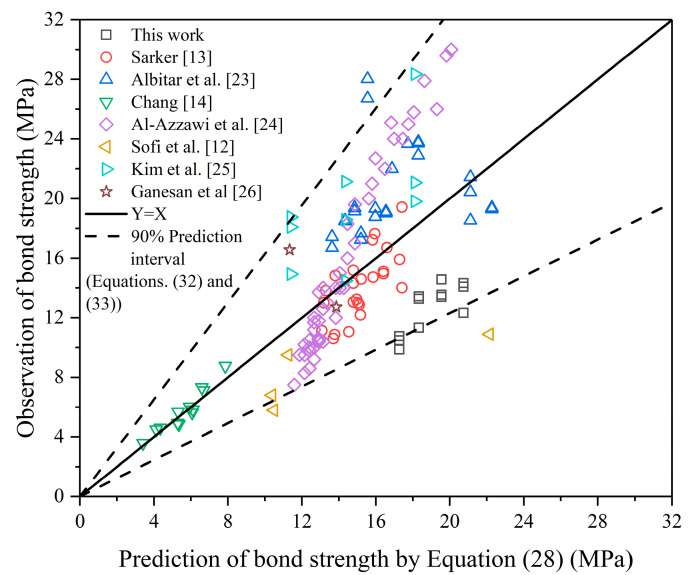
Comparison of the observation and prediction of bond strength by Equation (28).

**Figure 7 polymers-13-00875-f007:**
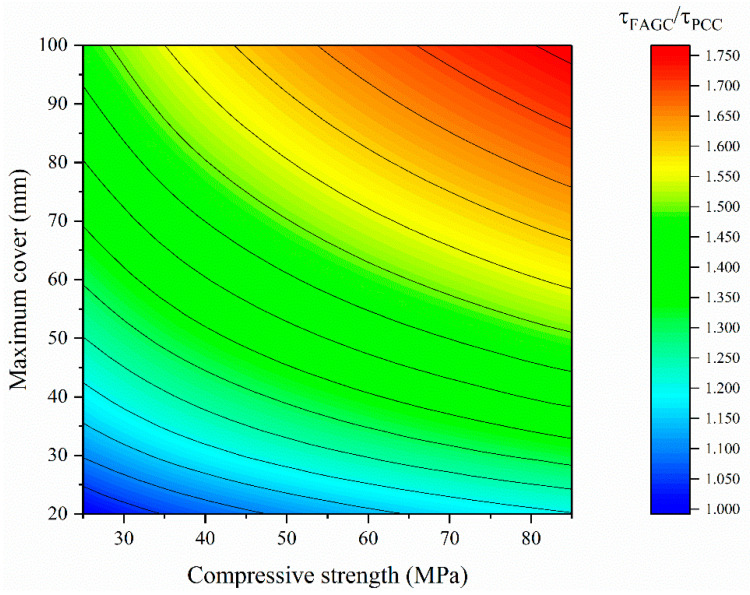
Comparison of bond strength of Portland cement concrete (PCC) and FAGC.

**Figure 8 polymers-13-00875-f008:**
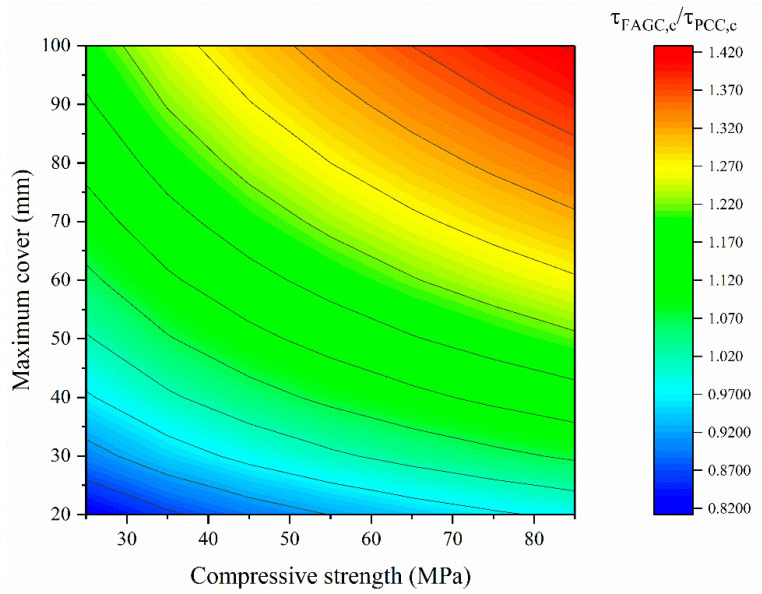
Comparison of the characteristic values of bond strength of PCC and FAGC.

**Figure 9 polymers-13-00875-f009:**
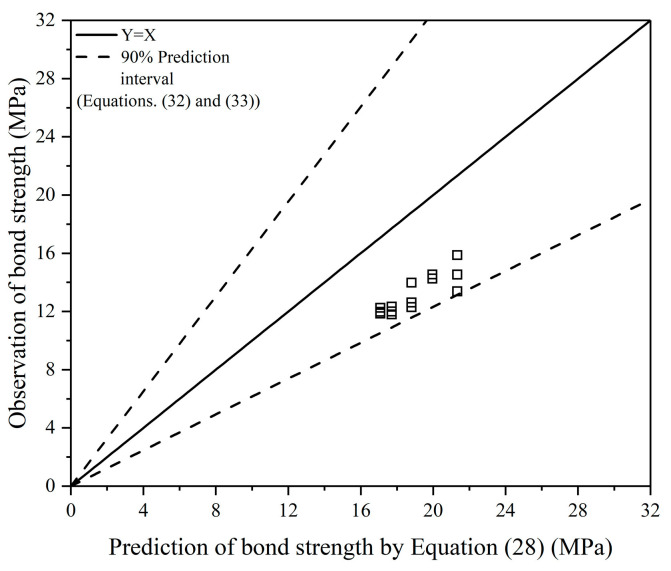
Validation of bond strength model.

**Figure 10 polymers-13-00875-f010:**
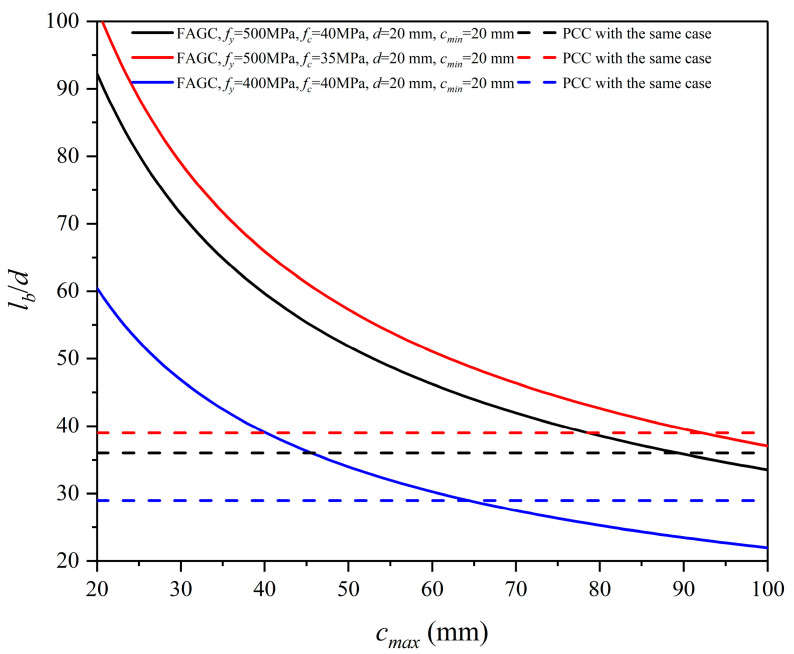
Design anchorage length of reinforced FAGC beams for some cases.

**Table 1 polymers-13-00875-t001:** Chemical composition of fly ash.

Composition	SiO_2_	Al_2_O_3_	Fe_2_O_3_	CaO	P_2_O_5_	Na_2_O	TiO_2_	MgO
Mass%	62.83	16.71	7.38	6.37	4.11	1.05	0.91	0.64

**Table 2 polymers-13-00875-t002:** Mix proportions and corresponding compressive strength.

Mix.	Coarse	Fine	Fly Ash	Alkaline Solution(kg/m^3^)			*f_c_*
	Aggregate (kg/m^3^)	Aggregate (kg/m^3^)	(kg/m^3^)	Na_2_SiO_3_	NaOH	Water	(MPa)
1	1173.00	527.00	500.00	142.86	18.00	39.14	58.51
2	1260.42	566.28	420.00	91.98	19.32	42.00	65.58
3	1250.28	561.72	420.00	112.00	19.64	36.36	75.90
4	1160.93	521.58	500.00	130.50	30.51	56.49	86.60
5	1240.14	557.16	420.00	130.50	19.98	32.22	69.87
6	1222.75	549.35	460.00	119.93	16.82	31.15	62.23
7	1211.64	544.36	460.00	110.40	28.17	45.43	92.86
8	1200.53	539.37	460.00	133.40	20.94	45.76	56.80
9	1185.08	532.43	500.00	121.67	23.28	37.55	79.83

**Table 3 polymers-13-00875-t003:** The conversion factors of compressive strength.

Strength Grade	100 mm Cube	150 mm Cube	100 × 200 mm^2^ Cylinder	150 × 300 mm^2^ Cylinder
C20-C40	0.762	0.8	-	1
C50	0.790	0.83		1
C60	0.819	0.86		1
C70	0.833	0.875		1
C80	0.848	0.89		1
	-	-	0.971	1

**Table 4 polymers-13-00875-t004:** The conversion factors of splitting tensile strength.

100 mm Cube	150 mm Cube	75 × 150 mm^2^ Cylinder	100 × 200 mm^2^ Cylinder	150 × 300 mm^2^ Cylinder
0.825	0.915	0.837	0.901	1

**Table 5 polymers-13-00875-t005:** Summary of splitting tensile strength and corresponding compressive strength of fly ash geopolymer concrete (FAGC, MPa).

Experiments	Specimens for *f_c_*	Specimens for *f_st_*	*f_c_* ^1^	*f_st_* ^1^	*f_c_^’^* ^1^	*f_st_^’^* ^1^
This work	100 mm	100 mm	58.51	2.826	46.25	2.335
	cube	cube	58.51	3.261	46.25	2.695
			58.51	3.306	46.25	2.732
			58.51	3.288	46.25	2.718
			58.51	3.722	46.25	3.076
			65.58	3.391	53.72	2.803
			65.58	3.137	53.72	2.592
			65.58	3.756	53.72	3.104
			65.58	3.363	53.72	2.78
			65.58	5.048	53.72	4.172
			75.9	4.057	63.25	3.353
			75.9	4.585	63.25	3.789
			75.9	4.075	63.25	3.368
			75.9	4.256	63.25	3.517
			75.9	4.76	63.25	3.934
			86.6	4.066	73.41	3.36
			86.6	4.691	73.41	3.877
			86.6	4.669	73.41	3.859
			86.6	4.46	73.41	3.686
			86.6	5.2	73.41	4.297
Gunasekera et al.	100 × 200 mm^2^	150 × 300 mm^2^	82.5	4.26	80.1	4.26
[16]	cylinder	cylinder	36.9	2.49	35.83	2.49
			29.6	1.6	28.74	1.6
			24.9	1.15	24.17	1.15
			86.5	4.67	83.98	4.67
			45.6	3.41	44.27	3.41
			36.5	2.32	35.44	2.32
			27.7	1.3	26.89	1.3
			87.4	4.72	84.85	4.72
			47	3.71	45.63	3.71
			42.8	2.9	41.55	2.9
			28.7	1.86	27.86	1.86
Olivia et al.	100 × 200 mm^2^	150 × 300 mm^2^	56.49	4.13	54.84	4.13
[17]	cylinder	cylinder	56.24	3.96	54.6	3.96
			60.2	4.29	58.45	4.29
			56.51	4.18	54.86	4.18
			58.85	4.1	57.14	4.1
			63.29	4.79	61.45	4.79
Ryu et al.	100 × 200 mm^2^	100 × 200 mm^2^	17.17	1.3	16.67	1.171
[15]	cylinder	cylinder	19.32	1.3	18.76	1.171
			19.46	1.53	18.89	1.378
			21.94	1.59	21.3	1.432
			24.14	1.97	23.44	1.775
			26.51	1.97	25.74	1.775
			26.67	2.18	25.89	1.964
			28.02	2.03	27.2	1.829
			29.35	2	28.5	1.802
			26.92	2.4	26.14	2.162
			30.01	2.22	29.14	2
			31	2.44	30.1	2.198
			33.98	2.68	32.99	2.414
			34.94	2.82	33.92	2.541
			35.27	2.73	34.24	2.459
			36.18	2.68	35.13	2.414
			56.02	3.65	54.39	3.288
			58.91	3.68	57.19	3.315
			61.19	3.93	59.41	3.541
Benny Joseph	150 mm	150 × 300 mm^2^	45	3.1	36	3.1
[18]	cube	cylinder	47	3.34	37.6	3.34
			56	3.45	46.48	3.45
			49	4.51	39.2	4.51
Sofi et al.	150 × 300 mm^2^	150 × 300 mm^2^	38.3	2.7	38.3	2.7
[19]	cylinder	cylinder	52.8	3.3	52.8	3.3
Aliabdo et al.	100 mm	75 × 150 mm^2^	22.5	2.45	17.14	2.059
[20]	cube	cylinder	27.5	2.75	20.95	2.311
			36.5	3.05	27.81	2.563
			35	3	26.67	2.521
			43.5	3.3	33.14	2.773
			13.5	1.7	10.29	1.429
			37.5	3.05	28.57	2.563
			33	2.7	25.14	2.269
			25.3	2.5	19.28	2.101
			31	3.1	23.62	2.605
Albitar et al.	100 × 200 mm^2^	100 × 200 mm^2^	18.66	2.04	18.12	1.838
[21]	cylinder	cylinder	33.17	3.08	32.2	2.775
			34.41	3.14	33.41	2.829
			29.45	2.96	28.59	2.667
			51.42	4.23	49.92	3.811
			53.42	5.55	51.86	5
			44.58	5.51	43.28	4.964
Hardjito	100 × 200 mm^2^	150 × 300 mm^2^	89	7.43	86.41	7.43
and Rangan	cylinder	cylinder	68	5.52	66.02	5.52
[22]			55	5.45	53.4	5.45
			44	4.43	42.72	4.43
Chang	100 × 200 mm^2^	150 × 300 mm^2^	37	3.62	35.92	3.62
[14]	cylinder	cylinder	30	2.96	29.13	2.96
			55	4.06	53.4	4.06
			48	4.48	46.6	4.48
			29	2.93	28.16	2.93
			51	4.65	49.51	4.65

^1^*f_c_* = experiment results of compressive strength, *f_st_* = experiment results of splitting tensile strength, *f_c_’* = the 150 × 300 mm cylinder compressive strength, *f_st_’* = the 150 × 300 mm cylinder splitting tensile strength.

**Table 6 polymers-13-00875-t006:** Summary of the bond strength and corresponding factors of FAGC.

Experiments	*τ* (MPa)	*f_c_* (MPa)	Specimens for *f_c_*	*f_c_^’^* (MPa)	*l*	*d*	*c_min_*	*c_max_*
Specimens for Bond Test					(mm)			
This work	11.34	65.58	100 mm	53.716	80	16	67	67
modified direct pull-out specimens	13.43	65.58	cube	53.716	80	16	67	67
	13.27	65.58		53.716	80	16	67	67
	14.56	75.9		63.247	80	16	67	67
	13.40	75.9		63.247	80	16	67	67
	13.54	75.9		63.247	80	16	67	67
	10.49	58.51		46.251	80	16	67	67
	10.77	58.51		46.251	80	16	67	67
	9.86	58.51		46.251	80	16	67	67
	14.08	86.6		73.407	80	16	67	67
	14.33	86.6		73.407	80	16	67	67
	12.34	86.6		73.407	80	16	67	67
P. K. Sarker. [13]	10.61	25.5	100 × 200 mm^2^	24.757	100	24	42	113
beam-end specimens	13.02	25.5	cylinder	24.757	110	24	44	113
	10.88	25.5		24.757	100	24	44	113
	13.82	25.5		24.757	120	24	65	113
	11.14	25.5		24.757	125	24	66	113
	14.83	25.5		24.757	110	24	64	113
	14.32	29.7		28.835	100	20	45	115
	13.05	29.7		28.835	100	20	45	115
	13.23	29.7		28.835	95	20	41	115
	15.19	29.7		28.835	110	20	64	115
	12.88	29.7		28.835	105	20	64	115
	11.07	29.7		28.835	115	20	66	115
	12.20	32.5		31.553	100	24	44	113
	14.59	32.5		31.553	100	24	45	113
	13.00	32.5		31.553	100	24	41	113
	14.72	32.5		31.553	100	24	63	113
	17.64	32.5		31.553	100	24	66	113
	17.24	32.5		31.553	100	24	62	113
	14.96	39.5		38.35	100	20	42	115
	15.12	39.5		38.35	100	20	42	115
	16.71	39.5		38.35	100	20	46	115
	19.42	39.5		38.35	100	20	68	115
	14.01	39.5		38.35	100	20	68	115
	15.92	39.5		38.35	100	20	64	115
M. Albitar et al. [23]	17.68	33	100 × 200 mm^2^	32.039	60	12	24	69
direct pull-out specimens	17.25	33	cylinder	32.039	60	12	24	69
	18.78	33		32.039	60	12	36	69
	19.33	33		32.039	60	12	36	69
	19.01	33		32.039	60	12	48	69
	19.12	33		32.039	60	12	48	69
	17.44	33		32.039	80	16	32	67
	16.71	33		32.039	80	16	32	67
	18.49	33		32.039	80	16	48	67
	18.52	33		32.039	80	16	48	67
	19.15	33		32.039	80	16	64	67
	19.37	33		32.039	80	16	64	67
	19.38	33		32.039	80	16	117	177
	19.29	33		32.039	80	16	117	177
	22.00	43		41.748	60	12	24	69
	23.68	43		41.748	60	12	36	69
	26.73	43		41.748	80	12	36	69
	28.02	43		41.748	80	12	36	69
	18.53	38		36.893	60	12	94	94
	20.44	38		36.893	60	12	94	94
	21.45	38		36.893	60	12	94	94
	23.70	38		36.893	80	16	92	92
	22.90	38		36.893	80	16	92	92
	23.79	38		36.893	80	16	92	92
Ee Hui Chang [14]	4.94	37	100 × 200 mm^2^	35.922	355	24	20	32
beam specimens	6.03	37	cylinder	35.922	303	20	28	32
	7.34	30		29.126	240	16	29	40
	5.63	55		53.398	356	24	25	28
	7.13	55		53.398	301	20	24	30
	8.77	48		46.602	243	16	28	40
	4.85	30		29.126	300	24	25	31
	4.50	29		28.155	452	24	23	27
	3.59	29		28.155	723	24	25	28
	5.84	48		46.602	300	24	24	27
	5.70	51		49.515	455	24	22	30
	4.61	51		49.515	722	24	24	30
Al-azzawi et al. [24]	9.50	17	100 × 200 mm^2^	16.505	80	16	72	72
direct pull-out specimens	10.20	18	cylinder	17.476	80	16	72	72
	10.40	21		20.388	80	16	72	72
	10.00	19		18.447	80	16	72	72
	12.00	20		19.417	80	16	72	72
	14.00	22		21.359	80	16	72	72
	10.40	22		21.359	80	16	72	72
	12.00	25		24.272	80	16	72	72
	14.00	26		25.243	80	16	72	72
	8.60	19		18.447	80	16	72	72
	10.00	20		19.417	80	16	72	72
	11.70	20		19.417	80	16	72	72
	9.20	20		19.417	80	16	72	72
	10.60	21		20.388	80	16	72	72
	12.60	22		21.359	80	16	72	72
	13.70	21		20.388	80	16	72	72
	14.00	25		24.272	80	16	72	72
	15.00	26		25.243	80	16	72	72
	7.50	16		15.534	80	16	72	72
	8.30	18		17.476	80	16	72	72
	9.80	19		18.447	80	16	72	72
	9.50	18		17.476	80	16	72	72
	10.50	19		18.447	80	16	72	72
	11.20	20		19.417	80	16	72	72
	10.50	21		20.388	80	16	72	72
	11.80	21		20.388	80	16	72	72
	13.00	23		22.33	80	16	72	72
	24.00	45		43.689	80	16	72	72
	22.00	39		37.864	80	16	72	72
	20.00	34		33.01	80	16	72	72
	26.00	58		56.311	80	16	72	72
	25.00	47		45.631	80	16	72	72
	24.00	42		40.777	80	16	72	72
	30.00	64		62.136	80	16	72	72
	29.60	62		60.194	80	16	72	72
	27.90	53		51.456	80	16	72	72
	17.00	30		29.126	80	16	72	72
	16.00	28		27.184	80	16	72	72
	14.00	27		26.214	80	16	72	72
	21.00	35		33.981	80	16	72	72
	19.60	30		29.126	80	16	72	72
	18.30	28		27.184	80	16	72	72
	25.80	49		47.573	80	16	72	72
	25.10	41		39.806	80	16	72	72
	22.70	36		34.951	80	16	72	72
Sofi et al. [12] beam-end specimens	9.50	30	150 × 300 mm^2^	30	168.8	12	36	106.5
	6.80	30	cylinder	30	216.8	16	48	106.5
	5.80	30		30	234	20	70	106.5
direct pull-out specimens	10.90	59.8		59.8	60	12	69	69
Kim, Jee Sang [25]	19.81	20	150 × 300 mm^2^	20	50	10	95	95
direct pull-out specimens	21.08	20	cylinder	20	50	10	95	95
	28.35	20		20	50	10	95	95
	18.58	20		20	80	16	92	92
	14.48	20		20	80	16	92	92
	21.14	20		20	80	16	92	92
	18.76	20		20	125	25	87.5	87.5
	14.93	20		20	125	25	87.5	87.5
	18.09	20		20	125	25	87.5	87.5
Ganesan et al. [26]	12.73	41.23	150 mm	32.984	100	12	69	69
direct pull-out specimens	16.57	41.23	cube□	32.984	150	16	67	67

**Table 7 polymers-13-00875-t007:** Models of splitting tensile strength of FAGC.

Equations	Basic Equation	Model	t-Value	*p* Value	*R^2^*
(20)	$f_{st}=kf_{c}^{n}$	$f_{st}=0.203f_{c}^{0.722}$	13.693	0.000	0.681
(21)	$f_{st}=a\left(0.556f_{c}^{0.5}\right)$	$f_{st}=0.880\left(0.556f_{c}^{0.5}\right)$	41.364	0.000	0.584
(22)	$f_{st}=aIn\left(f_{c}\right)-b$	$f_{st}=2.043ln\left(f_{c}\right)-4.417$	11.989	0.000	0.620
(23)	$f_{st}=a\left(2.329ln\left(f_{c}\right)-4.71\right)$	$f_{st}=0.808\left(2.329ln\left(f_{c}\right)-4.71\right)$	43.155	0.000	0.616

**Table 8 polymers-13-00875-t008:** Results of FAGC for validation of splitting tensile strength model (MPa).

Specimens for *f_c_*	Specimens for *f_st_*	*f_c_*	*f_st_*	*f_c_^’^*	*f_st_^’^*
100 mm	100 mm	69.87	4.086	57.23	3.371
cube	cube	69.87	3.848	57.23	3.175
		69.87	4.503	57.23	3.715
		69.87	4.330	57.23	3.572
		69.87	4.243	57.23	3.500
		62.23	4.128	50.97	3.406
		62.23	3.520	50.97	2.904
		62.23	3.333	50.97	2.750
		62.23	3.034	50.97	2.503
		62.23	3.966	50.97	3.272
		92.86	4.928	78.74	4.065
		92.86	4.889	78.74	4.033
		92.86	5.678	78.74	4.684
		92.86	5.967	78.74	4.923
		92.86	5.371	78.74	4.431
		56.80	4.073	44.87	3.360
		56.80	4.327	44.87	3.569
		56.80	3.226	44.87	2.662
		56.80	4.684	44.87	3.865
		56.80	4.284	44.87	3.534
		79.83	3.322	66.50	2.741
		79.83	3.563	66.50	2.939
		79.83	4.086	66.50	3.371
		79.83	5.609	66.50	4.628
		79.83	4.765	66.50	3.931

**Table 9 polymers-13-00875-t009:** Bond strength for validation of bond strength model.

Experiment	*τ* (MPa)	*f_c_* (MPa)	Specimens for *f_c_*	*f_c_^’^* (MPa)	*l*	*d*	*c_min_*	*c_max_*
Specimens for Bond Test					(mm)			
This work	13.98	69.87	100 mm	57.23	80	16	67	67
modified direct pull-out specimens	12.29	69.87	cube	57.23	80	16	67	67
	12.61	69.87		57.23	80	16	67	67
	11.79	62.23		49.194	80	16	67	67
	12.32	62.23		49.194	80	16	67	67
	11.99	62.23		49.194	80	16	67	67
	15.87	92.86		78.707	80	16	67	67
	14.53	92.86		78.707	80	16	67	67
	13.38	92.86		78.707	80	16	67	67
	12.25	56.8		44.896	80	16	67	67
	11.97	56.8		44.896	80	16	67	67
	11.86	56.8		44.896	80	16	67	67
	14.26	79.83		66.525	80	16	67	67
	14.54	79.83		66.525	80	16	67	67
	14.26	79.83		66.525	80	16	67	67

**Table 10 polymers-13-00875-t010:** Comparison of test and predicted cracking moment of reinforced FAGC beams.

Beam	*f_c_*	*b*	*h*	*M_cr_*	*M_cr_* Predicted by [37]	*M_cr_* Predicted by Equation (38)	Test/Prediction
	(MPa)	(mm)		(KN × m)	(KN × m)	(KN × m)	Ratio
GB1-1	37	200	300	13.40	10.39	12.68	1.06
GB1-2	42	200	300	13.55	10.86	13.90	0.97
GB1-3	42	200	300	13.50	10.61	13.90	0.97
GB1-4	37	200	300	14.30	9.66	12.68	1.13
GB2-1	46	200	300	15.00	11.65	14.84	1.01
GB2-2	53	200	300	16.20	12.27	16.44	0.99
GB2-3	53	200	300	16.65	12.02	16.44	1.01
GB2-4	46	200	300	16.05	10.91	14.84	1.08
GB3-1	76	200	300	19.00	15.13	21.33	0.89
GB3-2	72	200	300	20.00	14.43	20.51	0.98
GB3-3	72	200	300	21.00	14.18	20.51	1.02
GB3-4	76	200	300	19.90	14.39	21.33	0.93
						Average	1.00
				Standard Deviation			0.06

**Table 11 polymers-13-00875-t011:** Comparison of test and predicted crack spacing of reinforced FAGC beams.

Experiment	Beam	Length of Pure Bending Zone	Number of Cracks	*S_cr_*	Predicted *S_cr_*	Test/Prediction Ratio
		(mm)		(mm)	(mm)	
Sumajouw	GB1-2	1000	14	76.92	68.17	1.13
and Rangan [37]	GB1-3	1000	13	83.33	65.97	1.26
	GB3-1	1000	13	83.33	76.82	1.08
	GB3-2	1000	12	90.91	72.70	1.25
Kumaravel and	GPC-1	1000	14	76.92	96.99	0.79
Thirugnanasambandam	GPC-2	1000	10	111.11	96.99	1.15
[40]					Average	1.11
				Standard Deviation		0.17

**Table 12 polymers-13-00875-t012:** Details of the tested reinforced FAGC beams.

Experiment	Beam	*f_c_*	*d*	*c_max_*	*c_min_*	$\sum^{\;}O_{r}$	*A_t_*
		(MPa)	(mm)	(mm)	(mm)	(mm)	(mm^2^)
Sumajouw	GB1-2	42	16	25.5	25	150.72	129.52
and Rangan [37]	GB1-3	42	20	25	22.5	188.4	155.65
	GB3-1	76	12	28.5	25	113.04	88.94
	GB3-2	72	16	25.5	25	150.72	112.51
Kumaravel and	GPC-1	46.61	16	20	10.25	138.16	114.68
Thirugnanasambandam [40]	GPC-2	46.61	16	20	10.25	138.16	114.68

## Data Availability

The data presented in this study are available on request from the corresponding author.

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
