# Peer review of "Practical Prediction Models of Tensile Strength and Reinforcement-Concrete Bond Strength of Low-Calcium Fly Ash Geopolymer Concrete"

_polymers, 2021, doi:10.3390/polym13060875_

Round 1
Reviewer 1 Report
The work is potentially of interest for the journal readership.
However, the current version of the manuscript cannot be accepted in the present form. It need a thorough revision of English and a check of formal aspects and scientific writing style. Th advice of either a native speaker or a professional proofreader is probably essential in this task.
- both English and scientific writing need to be revised/improved; - several references do not work ("Error! Refer-39 ence source not found."); - the title is too long; - Table 2: it would be advisable putting units in the table (rather than in the caption); - eq.1: please, check whether l or l' is the relevant parameter (report it in figure 2 and 3, too); - as far as one can understand, the conversion factors reported in Table 3 and 4 have been calibrated for concrete; here, the authors employ them for geopolymer: is there an experimental basis for this assumption? The authors should clarify this aspect. - table 7: the resulting values of R2 do not seem sufficiently high: which are the corresponding values achieved for similar calibration on concrete? Are the former acceptable? The authors should further comment on this; - the conclusion section should be rearranged: after an introductory sentence, the main findings have to be listed in concise items. The authors are requested not to refer to mere empirical evidence; instead, they should try to point out the mechanical reasons behind them.Author Response
General Comments: The work is potentially of interest for the journal readership.
However, the current version of the manuscript cannot be accepted in the present form. It need a thorough revision of English and a check of formal aspects and scientific writing style. The advice of either a native speaker or a professional proofreader is probably essential in this task.
Response: Thank you for your comments. We have made every effort to improve the quality and clarity of the language and scientific writing throughout the manuscript. The details are given in response to specific comments below.
Comment 1: both English and scientific writing need to be revised/improved;
Response 1: We have made every effort to improve the quality and clarity of the language and scientific writing throughout the manuscript. The Abstract, Introduction and Conclusions are rewritten.
Comment 2: several references do not work ("Error! Reference source not found.");
Response 2: Now the reference numbers can work correctly.
Comment 3: the title is too long;
Response 3: The title has been changed to “Practical prediction models of tensile strength and reinforcement-concrete bond strength of FAGC”.
Comment 4: Table 2: it would be advisable putting units in the table (rather than in the caption);
Response 4: The text has been modified as requested.
Comment 5: eq.1: please, check whether l or l' is the relevant parameter (report it in figure 2 and 3, too);
Response 5: “l” is the relevant parameter and the “,” has been removed. We also marked “l” in Figure 2.
Comment 6: as far as one can understand, the conversion factors reported in Table 3 and 4 have been calibrated for concrete; here, the authors employ them for geopolymer: is there an experimental basis for this assumption? The authors should clarify this aspect.
Response 6: The larger the volume of the concrete subjected to stress, the more likely it is to contain an element of a given low strength. As a result, the measured strength of the specimen decreases with an increase in its size. Due to the friction between the specimen and the testing machine, the measured compressive strength of the cube specimen is higher than that of the cylinder specimen with the length-to-diameter ratio of 2.
So far, no study has investigated the effect of size and shape of tested specimens on strength for FAGC, but the above two conclusions still apply to FAGC. Therefore, in this work, we suppose that the conversion factors of compressive strength and splitting tensile strength for PCC also apply to FAGC. This assumption has also been used in reference [28].
Comment 7: table 7: the resulting values of R2 do not seem sufficiently high: which are the corresponding values achieved for similar calibration on concrete? Are the former acceptable? The authors should further comment on this;
Response 7: Although the resulting values of R2 (around 0.6) in table 7 do not seem sufficiently high, the R2 corresponding to the P value of 0.01 is only 0.07 when there are 90 data points [33]. The P value < 0.01 shows that there is a highly significant linear relationship between the independent variable X and the dependent variable Y. Therefore, Eqs. (20-23) can describe the relationship between splitting tensile strength and compressive strength of FAGC satisfactorily.
Comment 8: the conclusion section should be rearranged: after an introductory sentence, the main findings have to be listed in concise items.
Response 8: We have rearranged the Conclusions section, as follows:
“This study has established the databases of splitting tensile strength and bond strength of FAGC, developed and verified the prediction models and the corresponding prediction intervals of splitting tensile strength and bond strength of FAGC, and used the strength models to calculate the design anchorage length and estimate the cracking moment, crack spacing and width of reinforced FAGC beams.
Based on this study, the following conclusions are obtained:
- Compared with the previous strength models of FAGC, the tensile strength model in this study considers the effect of shape and size of tested specimens on strength, as well as the bond strength model in this study considers the cover to diameter ratio and the diameter to development length ratio. Therefore, the models in this study can be used as the design equations for estimating the tensile strength and reinforcement-concrete bond strength of FAGC.
- The strength models provide the corresponding 90% prediction intervals. The lower limit of the prediction intervals is the characteristic value of the strength.
- The splitting tensile strength of FAGC is slightly lower than that of PCC with the same compressive strength, while the scatter of the splitting tensile strength of FAGC is close to that of PCC.
- The scatter of the bond strength of FAGC is larger than that of PCC. This results in the fact that for the bond strength of FAGC, although the estimated mean value is higher than that of PCC in the same case, the characteristic value may be lower than that of PCC in the case of a small bar spacing.
- The strength prediction models of PCC cannot be used for FAGC.
- To ensure adequate anchorage and suitable design anchorage lengths of reinforced FAGC beams, the minimum bar spacing needs to be restricted in the design code for FAGC.
- Incorporating the models into the prediction models of the cracking behaviors for PCC gives good predictions on the cracking moment and crack spacing of reinforced FAGC beams.”
Comment 9: The authors are requested not to refer to mere empirical evidence; instead, they should try to point out the mechanical reasons behind them.
Response 9: We fully agree with this view. In this study, we did not simply measure the strength of FAGC, but further proposed the practical strength models which can be used as the design equations for FAGC.

Reviewer 2 Report
The manuscript presents an interesting study conducted on the fly ash based geopolymers, in order to develop a tensile strength model considering different parameters (compressive strength, bond strength, splitting tests, shape and size of the specimens). However, multiple affirmations aren’t supported by the provided references or the experimental results obtained, and few other issues must be addressed. Therefore, the paper needs minor revisions before it is processed further, some comments follow:
Line 45: The term "reinforced” is redundant in the "reinforced concrete members”, (concrete members refers only to reinforced concrete) please remove it.
Line 45-46: "The most important application of concrete in construction is reinforced concrete 45 members.” Please provide experimental data or cite previous studies to support you affirmation.
Lines 53-55: The following sentence is unclear: " However, the tensile strength and reinforcement-concrete bond strength of FAGC should be incorporated into the prediction models for PCC, since there are differences between those of FAGC and PCC.” If there are differences between those two (as supported in lines 56-61, and line 286), how can be this incorporated? Please cite previous studies to support your affirmation. Please enhance the clarity of the sentence.
Line 178: Please replace the term “height” with "length”.
Author Response
General Comments: The manuscript presents an interesting study conducted on the fly ash based geopolymers, in order to develop a tensile strength model considering different parameters (compressive strength, bond strength, splitting tests, shape and size of the specimens). However, multiple affirmations aren’t supported by the provided references or the experimental results obtained, and few other issues must be addressed. Therefore, the paper needs minor revisions before it is processed further, some comments follow.
Response: Thank you for your comments. We have revised the manuscript to address the matters raised. The details are given in response to specific comments below.
Comment 1: Line 45: The term "reinforced” is redundant in the "reinforced concrete members”, (concrete members refers only to reinforced concrete) please remove it.
Response 1: The text has been modified as requested.
Comment 2: Line 45-46: "The most important application of concrete in construction is reinforced concrete members.” Please provide experimental data or cite previous studies to support you affirmation.
Response 2: We have been added Reference [9] to support the affirmation. The reference states “The most important application of concrete in building construction is nonetheless reinforced concrete structural members.” in its abstract.
Comment 3: Lines 53-55: The following sentence is unclear: " However, the tensile strength and reinforcement-concrete bond strength of FAGC should be incorporated into the prediction models for PCC, since there are differences between those of FAGC and PCC.” If there are differences between those two (as supported in lines 56-61, and line 286), how can be this incorporated? Please cite previous studies to support your affirmation. Please enhance the clarity of the sentence.
Response 3: The sentence has been changed to “To estimate the cracking behaviours and calculate the design anchorage length, it is vital to accurately estimate the tensile strength and reinforcement-concrete bond strength from the compressive strength using the specific strength models of FAGC rather than the strength models of PCC.”. This sentence emphasizes the importance of developing the specific strength models of FAGC.
Comment 4: Line 178: Please replace the term “height” with "length”.
Response 4: The text has been modified as requested.

Reviewer 3 Report
The present study reports the prediction models of tensile strength and reinforcement concrete bond strength of FAGC. There are following questions:
- Abstract need to be rewritten to report about the main and new findings obtained in this paper briefly.
- It needs to be clearly stated the contributions and novelty of the manuscript in the introduction section.
- The author should compare the results with others’ research to confirm the contribution of this research.
- Many "Error! Reference source not found" appear in the manuscript.
- Conclusion is too long, should be described point by point and more precise and clearly.
Author Response
General Comments: The present study reports the prediction models of tensile strength and reinforcement concrete bond strength of FAGC.
Response: Thank you for your comments. We have revised the manuscript to address the matters raised. The details are given in response to specific comments below.
Comment 1: Abstract need to be rewritten to report about the main and new findings obtained in this paper briefly.
Response 1: The Abstract has been rewritten as requested, as follows:
“There have been a few attempts to develop prediction models of splitting tensile strength and re-inforcement-concrete bond strength of FAGC (low-calcium fly ash geopolymer concrete), however, no model can be used as a design equation. Therefore, this paper aimed to provide practical pre-diction models. Using 115 test results for splitting tensile strength and 147 test results for bond strength from the experiments and previous literature, considering the effect of size and shape on strength and structural factors on bond strength, this paper developed and verified updated pre-diction models and the 90% prediction intervals by regression analysis. The models can be used as the design equations and applied for estimating the cracking behaviors and calculating the design anchorage length. The strength models of PCC (Portland cement concrete) overestimate the splitting tensile strength and reinforcement-concrete bond strength of FAGC, so PCC’s models are not recommended as the design equations.”.
Comment 2: It needs to be clearly stated the contributions and novelty of the manuscript in the introduction section.
Response 2: We have added the Research Significance Section in Introduction to show the paper’s specific original contribution more clearly.
Research Significance:
“At present, there is no design equation developed to predict tensile strength and reinforcement-concrete bond strength of FAGC as functions of compressive strength, which hinders the use of FAGC for large-scale field applications. Reliance on the existing design equation for PCC may lead to unsafe structural design. So far, a few studies have proposed prediction models for FAGC; however, there are limitations in those prediction models as shown in Section 1.1. This work addresses these knowledge gaps, provides practical prediction models of tensile strength and reinforcement-concrete bond strength of FAGC, and uses the strength models to estimate the cracking behaviours of reinforced FAGC beams and calculate the design anchorage length.
In addition, it is noting that the data is from FAGC with different sources of fly ash and different mix designs, so the models do not depend on the sources of fly ash and the mix designs.”.
Comment 3: The author should compare the results with others’ research to confirm the contribution of this research.
Response 3: We fully agree with this view. And we emphasize this view in the Introduction and Conclusions.
In the Introduction, we list the limitations of the previous models in others’ research, as follows:
“In previous attempts to develop the prediction models of the splitting tensile strength of FAGC, comparatively few data have been used considering the large scatter [15,16,21]. In addition, the specimens for strength test in different studies were different, but the effect of the shape and size of the tested specimens on strength was not considered in previous attempts [15,16,21].”;
“in these previous attempts for developing the bond strength model, only the effect of compressive strength was considered, and the effects of structural factors, viz, the cover c, bar diameter d, development length l, on bond strength were ignored [23,24].”.
In the Conclusions, we emphasize the difference between the models in this study and previous models in others’ study, as follows:
“Compared with the previous strength models of FAGC, the tensile strength model in this study considers the effect of shape and size of tested specimens on strength, as well as the bond strength model in this study considers the cover to diameter ratio and the diameter to development length ratio. Therefore, the models in this study can be used as the design equations for estimating the tensile strength and reinforcement-concrete bond strength of FAGC.”.
In additions, compare with others’ research, this study also investigates the characteristic value of bond strength, and concludes that the characteristic value of bond strength of FAGC may be lower than that of PCC in the case of a small bar spacing. It can be also seen in the Conclusions.
Comment 4: Many "Error! Reference source not found" appear in the manuscript.
Response 4: Now the reference numbers can work correctly.
Comment 5: Conclusion is too long, should be described point by point and more precise and clearly.
Response 5: We have rearranged the Conclusions section. The words number of Conclusions has decreased from 411 to 330. And conclusion has been described point by point.
The Conclusions are as follows:
“This study has established the databases of splitting tensile strength and bond strength of FAGC, developed and verified the prediction models and the corresponding prediction intervals of splitting tensile strength and bond strength of FAGC, and used the strength models to calculate the design anchorage length and estimate the cracking moment, crack spacing and width of reinforced FAGC beams.
Based on this study, the following conclusions are obtained:
- Compared with the previous strength models of FAGC, the tensile strength model in this study considers the effect of shape and size of tested specimens on strength, as well as the bond strength model in this study considers the cover to diameter ratio and the diameter to development length ratio. Therefore, the models in this study can be used as the design equations for estimating the tensile strength and reinforcement-concrete bond strength of FAGC.
- The strength models provide the corresponding 90% prediction intervals. The lower limit of the prediction intervals is the characteristic value of the strength.
- The splitting tensile strength of FAGC is slightly lower than that of PCC with the same compressive strength, while the scatter of the splitting tensile strength of FAGC is close to that of PCC.
- The scatter of the bond strength of FAGC is larger than that of PCC. This results in the fact that for the bond strength of FAGC, although the estimated mean value is higher than that of PCC in the same case, the characteristic value may be lower than that of PCC in the case of a small bar spacing.
- The strength prediction models of PCC cannot be used for FAGC.
- To ensure adequate anchorage and suitable design anchorage lengths of reinforced FAGC beams, the minimum bar spacing needs to be restricted in the design code for FAGC.
- Incorporating the models into the prediction models of the cracking behaviors for PCC gives good predictions on the cracking moment and crack spacing of reinforced FAGC beams.”
